# “Enticing” but Not Necessarily a “Space Designed for Me”: Experiences of Urban Park Use by Older Adults with Disability

**DOI:** 10.3390/ijerph18020552

**Published:** 2021-01-11

**Authors:** Meredith Perry, Lucy Cotes, Benjamin Horton, Rebecca Kunac, Isaac Snell, Blake Taylor, Abbey Wright, Hemakumar Devan

**Affiliations:** 1Centre for Health, Activity and Rehabilitation Research (CHARR), School of Physiotherapy, University of Otago, Wellington 6021, New Zealand; hemakumar.devan@otago.ac.nz; 2School of Physiotherapy, University of Otago, Wellington 6021, New Zealand; Lucycotes14@gmail.com (L.C.); Horty.ben@hotmail.co.nz (B.H.); Rebecca.kunac@gmail.com (R.K.); Isaacsnell13@gmail.com (I.S.); Blaket.97@gmail.com (B.T.); Abbey.wright@outlook.co.nz (A.W.)

**Keywords:** accessibility, disability, green spaces, urban parks, public health, older adults

## Abstract

Urban parks are spaces that can enhance older adults’ physical, social and psychological wellbeing. As the prevalence of older adults with disability increases, it is important that urban parks are accessible to this population so that they too might gain health benefits. There is limited literature investigating the experiences of urban parks by older adults with disability. This qualitative study, set in a region of New Zealand, explored the experiences, including accessibility, of urban parks by 17 older adults (55 years and older) with self-reported disabilities. Three focus groups (*n* = 4, 5 and 4 people) and four individual interviews were undertaken. Data were analyzed using the General Inductive Approach. Two primary themes of “Enticing” and “Park use considerations” are presented. Urban parks and green spaces are perceived to provide an environment for older adults with a disability to improve their physical, psychosocial and spiritual health, and social connectedness. Parks that are not age, ability or culture diverse are uninviting and exclusive. Meaningful collaboration between park designers, city councils and people with disability is required to maximize the public health benefits of parks and make parks inviting and accessible for users of all ages, cultures and abilities. Park co-design with people with disability may provide one means of improving accessibility and park usability and thus park participation by older adults with disability.

## 1. Introduction

Internationally, the number of older adults is increasing and estimates suggest there will be 2.1 billion older adults by 2050 [1]. Approximately 15% of the world’s population (over 1 billion people) live with some form of disability [2]. Disability can be considered as the imposed limitations people with impairments experience in society. Impairments include physical, sensory, neurological, psychiatric or psychological, intellectual or other impairments [3]. These restrictions can negatively affect activities and participation. The prevalence of disability increases with age. In 2013, the number of people aged over 65 in New Zealand was 12.3% and this number is projected to double (26.7%) by 2063 [4]. Almost 60% of this population reported living with a disability (inability to use one or more extremities, decreased strength to walk, grasp, or lift objects and/or requiring mobility aids) [5]. However, the prevalence of disability in the general population is 25%. This phenomenon is not unique to New Zealand. The United Nations estimates that over 45% of older adults (aged 60 years and over) worldwide likely live with and are impacted by disability [6]. Accumulation of health risks across a lifespan of disease, injury, and chronic illness lead to this higher prevalence of disability amongst older adults [6].

Physical activity plays an important role in maintenance of health and wellbeing for older adults [7] and likewise for people with disability [8,9,10]. A very recent review of reviews and meta analyses [11] suggests older adults who attain sufficient levels of daily physical activity reduce their functional decline and reduce risk for dementia (including Alzheimer’s), depression, cancers, fractures, recurrent falls and cardiovascular disease. They have better ageing trajectories with improved ability to complete activities of daily living, better quality of life, and improved cognitive function. A sufficient daily level of physical activity has also been shown to positively affect physical health [12], helping to reduce high blood pressure, diabetes, heart disease, cholesterol levels, mortality and morbidity [13], and improve and sustain bone, joint and muscle health, which has been shown to reduce arthritic pain [14]. When physically active, older adults have been shown to report fewer feelings of stress, loneliness and anxiety compared to others their age [15].

Yet older adults, when compared to the rest of the population, are less likely to engage in sufficient physical activity to meet recommended daily guidelines [7,16] and are more likely to live sedentary lifestyles [15]. For older adults living with disability, meeting daily physical activity guidelines can be even more difficult [9,17,18]. One review considered the barriers and facilitators to physical activity participation amongst people with physical disabilities and reported numerous barriers across the social ecological model at all levels (intrapersonal to policy) in the health care and recreation sectors [9].

Urban parks and public green spaces can play a vital role in facilitating physical activity [14,19] and because they are usually low or no-cost they can theoretically provide a space for diverse populations, including age and ability, to meet recommended daily guidelines for physical activity [13,16,20]. Such spaces encompass (a) local neighborhood parks that provide for a community’s social and recreational requirements; (b) community parks that serve several neighborhoods and (c) destination parks that are premiere parks [21,22] that people across the city travel to. Urban parks may include benches, picnic or fishing areas, walking paths, trees, flowerbeds, wild gardens, play equipment or waterways [21] but amenities such as cafés and bathroom facilities are typically only present in the larger community or destination parks.

People use parks for exertive sports such as organized competitive games, individual pursuits such as jogging or (dog) walking, or as a place for yoga or Tai Chi [14]. Urban parks also provide a space for relaxing, appreciating nature, music and art and connecting with friends and family [14]. Older adults who access parks often meet the recommended physical activity levels of 30 min of moderate intensity or 15 min of vigorous intensity exercise, 5 days per week [12,23]. Recent research suggests that the link between urban parks and meeting recommended physical activity guidelines may be related to individual factors such as education, gender and employment status [23]. Nevertheless, early studies exploring park use with older adults, found that half of older adult participants reported being in a better mood after doing light-moderate exercise in a park [24], with use of parks and green spaces improving social integration and strengthening social ties, which are predictors of wellbeing and longevity in this age group [25]. More recently, Veitch et al. [26] reported that older adults value parks for walking, sitting/relaxing, enjoying nature and socializing with others. Furthermore, Sales et al. [27] reported that older adults who participated in a novel exercise program in a park designed specifically for older adults enjoyed the program and reported improved physical and psychological benefits, along with improved ability to undertake activities of daily living.

Despite the assumption that urban parks are available for all, literature suggests such spaces are often not necessarily accessible for people living with disability [28,29,30]. Therefore, given the known health benefits and opportunity for social connectedness that urban parks can provide, it seems imperative to understand the experience of park use by older adults with disability. However, to date, there is a paucity of research exploring this intersecting area. The objective of this study was to understand the experience of urban park use and accessibility by older adults living with disability, in the Greater Wellington region of New Zealand. The study aimed to interpret the positive and negative experiences of park use as well as facilitators and barriers to park use and accessibility.

## 2. Materials and Methods

### 2.1. Study Design

This manuscript reports the qualitative component of a sequential mixed-methods study (QUAN → QUAL) [31] investigating accessibility and usability of urban parks in the Greater Wellington Region of New Zealand. The survey of 1000 older adults randomly selected from the electoral roll compared park use by older adults with and without disability (Author, manuscript in preparation). The qualitative component of this study was conducted subsequent to the survey and its intent was to generate a better understanding of the survey results by exploring the experiences and reasons for urban park use in a selected group of older adults with disability. Reporting meets the COnsolidated criteria for REporting Qualitative research (COREQ) guideline [32]. The COREQ guideline is comprised of 3 Domains (Research team and reflexivity, Study design, and Analysis and findings) totaling 32 items. The demonstration of each item on the guideline, within a manuscript, ensures that minimal reporting standards for qualitative studies are met. The COREQ was developed by the Enhancing the QUAlity and Transparency Of health Research (EQUATOR) network. Ethical approval for the mixed-methods study was gained from the University of Otago Human Ethics Committee (Health) (Ref number H16/118).

### 2.2. Participants and Recruitment

Older adults (aged 65 years and over) with self-reported disability who participated in the survey and who resided in the Greater Wellington Region (which has a population of over 500,000 people) were recruited for this study. To facilitate comparison with already known New Zealand data, disability was defined as per the 2013 New Zealand Disability survey, as a long-term limitation (resulting from impairment) in a person’s ability to carry out daily activities [33]. Participants with self-reported disability indicated their willingness to be interviewed as part of their survey response. To ensure diversity of viewpoints, purposive sampling was used, paying attention to the inclusion of relevant variations in perceived health status, physical activity levels, ethnicity and type of self-reported disability [34]. A total of 87 people (from the 324 responders with disability) indicated a willingness to be interviewed on their returned survey. Potential interviewees were contacted by telephone or email, provided with an information sheet and a consent form, and invited to attend a focus group on a fixed date. Twelve participants were recruited via this method. Due to a limited number of responses from specific ethnicities and from one geographical area of the Greater Wellington region, two additional recruitment strategies were undertaken. A disability advisory group was asked to advertise the study to its members in the aforementioned area and already recruited participants were asked to approach older adults they knew with disability (snowballing). These strategies resulted in another 5 participants being recruited. Two men recruited by this method identified as Māori and Pacific but were 55 years of age. Due to longstanding health inequities in New Zealand, there is a difference of over 7 years life expectancy and a higher adjusted prevalence of disability for these two ethnic groups [33,35]. Therefore, a conscious decision to include these two participants was made. All participants provided written consent to participate in the study.

### 2.3. Data Collection

Data were collected via three focus groups and four individual interviews (for those unexpectedly unable to attend the focus group meetings) with a hand held audio recording device. A semi-structured interview guide was developed and used to generate open, flexible discussion in the focus groups and individual interviews (Table 1). The schedule contained core questions and potential prompts were used to encourage richer detail on specific discussion [36,37]. However, due to the semi-structured nature of the interviews, time spent on each question differed between each interview due to the group dynamics. Furthermore, questions were not necessarily asked and answered in a specific order.

The lead researcher (MP), who has over 10 years of qualitative research experience, led the focus groups whilst another (HD) took field notes. MP or HD both led two individual interviews, independently. Field notes were completed after each individual interview by the respective researcher and recorded initial impressions from the interview, the facial expressions and emphasis of the participants and any additional comments that arose after the recording device had been turned off. All audio recordings were transcribed verbatim by an independent transcriber. The focus groups ranged from 49–86 min in duration and individual interviews ranged from 17–33 min. Two focus group sessions were conducted in community/council owned buildings and one in a university building. Individual interviews were undertaken at a time and place that suited the participants (local park, community buildings and in a quiet room at the university). Demographic information including age, education, income, and ethnicity were collected.

### 2.4. Data Analysis

The General Inductive Approach was used to analyze interview data [36] and NVivo 10 [38] was used to assist thematic organization. This stage is systematic and pragmatically guided by the research aims [36] with the intent to build themes that represent the nuances that arise from varied responses around a particular phenomenon [36]. While the objectives of the research frame the analysis, concepts or theories may be explored deductively; it is accepted that the results are reflective of the participants’ experiences but also the researchers’ interpretation of the phenomenon [36]. The researchers in this project were all physiotherapists, with the lead researcher having over 20 years of clinical experience.

Two groups of three coders, (LC, BH, RK and IS, BT AW) used independent parallel coding [36] to inductively code raw transcript data. This included the field notes from the interviews. Analysis involved reading the transcripts multiple times and identifying participant text that answered the research aim. Test segments were labelled with a code which endeavored to describe the essence of what the participant had talked about in relation to park use experience and accessibility. The coding groups then swapped to compare their codes. Codes were collated, then grouped into categories. Discussion with the whole research team, at three separate time points, enabled categories superfluous to the research objectives to be removed or collapsed (to reduce overlap) until a coherent understanding of the results was reached.

Then, organization of the categories was contextualized within the theoretical perspective of the social model of disability and the barriers to access created by society (i.e., social-ecological model) [3,39]. These models are of particular interest to the lead researcher (MP). The term accessibility was also of relevance to the interpretation of the data. For the purposes of this research accessibility was considered as per Article 9—Accessibility of the United Nations Conventions on the Rights of Persons with Disabilities,


*To enable persons with disabilities to live independently and participate fully in all aspects of life, States Parties shall take appropriate measures to ensure to persons with disabilities access, on an equal basis with others, to the physical environment, to transportation, to information and communications, including information and communications technologies and systems, and to other facilities and services open or provided to the public, both in urban and in rural areas *[39]*.*


The whole research team including senior researchers met to finalize and reorganize the sub-themes and themes and write descriptions for each. A summary of the results was sent to participants to ensure final themes reflected their experiences. Three participants responded expressing their enjoyment of being involved but no changes were made to the results following this process. Perry et al. [40] demonstrate the highly iterative and structured process of this type of analysis.

### 2.5. Trustworthiness of Analysis

A semi-structured interview approach was chosen as it allows for gathering of information and opinions from an array of individuals, creating strong and reoccurring viewpoints throughout the interviews. This approach provides some guidance while inviting collaboration of ideas, securing unique insights from group interaction, and allowing for identification of topics most important to participants [41]. The independent parallel coding method enabled consistency within the findings. Members who coded the data kept reflective statements and an audit trail to ensure dependability and confirmability of data analysis [42], and in effort to acknowledge researcher bias [43]. This involved recording key summary points during our analysis, decisions made by our research team and re-organization of initial groupings into sub-themes and themes. Verification of the results occurred via member checking [44]. Finally, quotes that exemplify interpretation of the data were included. Quotes from the Focus Group held in North East, Greater Wellington region were labelled (FG1); from North West, Greater Wellington region (FG2); from South, Greater Wellington region (FG3); and individual interviews numbered consecutively at P1 to P4.

## 3. Results

A total of 17 older adults with disability were interviewed over three focus groups in the Greater Wellington Region; South (*n* = 4 participants), North East (*n* = 5 participants) and North West (*n* = 4 participants). Four additional participants were individually interviewed. Disabilities reported include: osteoarthritis, persistent pain, multiple sclerosis, double amputation, depression, autism spectrum disorder, vision impairment and deafness. Table 2 provides a summary of participant demographic data.

Two primary themes were constructed. These were:Enticing—the value of parks to wellbeing.Park use considerations—experiences of navigating an accessible journey, the park environment, and contending with exclusive park design.

Table 3 shows these, with their associated sub-themes and their descriptions. They provide a context for understanding the experience and accessibility of urban parks for older adults with disability.

### 3.1. Enticing

Two core reason as to why parks were enticing were evident in discussions with participants. Parks enabled connection and participants valued perceived health benefits. Participants did not necessarily talk about the neighborhood or community parks that were situated geographically closest to them, but instead, talked more generally about their experiences with urban parks.

#### 3.1.1. Connecting of ‘Self’

Parks were enticing because they allowed participants to connect to the outside world. This was via an increased awareness of their ‘self’ as a part of nature; to significant others such as family and friends; and to live vicariously through watching other park users. Parks provided a place for participants to physically and metaphorically ground themselves; to consider nature and themselves as a part of nature. Participants became more aware of their ‘self’ in the park environment and talked about their experience from a sensory perspective. They described *“the smell of the flowers and the color” (FG1)* from the gardens, bush and noise from the birds, and used emotive language such as, *“I love the scenery” (P2).* The opportunity to re-connect provided a relief from the busyness and noise of an urban environment, *“Parks are very important I think. … Especially round the urban environment” (FG2).* Connecting to self, typically required space that was welcoming and did not necessarily involve physical activity, *“Somewhere to sit and something to look at” (FG1).*

Participants also discussed the enjoyment they got from being at the park with their family and friends, *“Being involved with not only my family, but I also get to meet friends, and other people” (FG2)* and described inter-generational park experiences, *“[I] go with my wife, three children and four grandchildren” (FG1).* They talked about going to the playground, picnics, walks and watching sport with significant others. These moments in time were poignant and participants recognized their value, with parks providing a ‘safe’ place to enjoy being with others and making these connections, as well as creating new memories.


*“When you all go out in an environment, you talk about what you’re looking at, and you’re doing it together. For me, it’s my ultimate enjoyment.—I love doing anything with my granddaughters, they’re my life ... spending that amount of time, together; interacting and enjoying looking at the same things. Enjoying everything, conversation, and just being there together in such a worthwhile environment. Parks are it for that” (FG2).*


Some participants talked about connecting with others they did not know and furthermore enjoying and appreciating the opportunity to do so.


*“Because you sit down and you talk to people ... You know, you’ll sit down, and you’re sitting next door to me, and I’ll say ‘nice day,’ and the conversation flows from there ... like it’s that interaction with other people [I like].” (FG2).*


When people they did not know initiated conversation with them the interaction was still appreciated as it helped to reinforce self-worth and identity. In a park environment, participants perceived that they were considered to be trustworthy and worth engaging with because they were older adults.


*“And sometimes people will talk to you! Especially if you’re elderly, I notice people will often come to you more than when we were younger. And you know, they’ll see you sitting down, and they’ll often stop and talk.” (FG3).*


Finally, participants reported that there was always activity in a park. This enabled the vicarious enjoyment from watching others, *“Us oldies like that park too, because we can sit…have a cup of coffee and watch the children playing” (P2).* These moments provided a space for participants to remember past experiences, *“I love watching [the sport], cos I used to play a lot of sports, rugby, volleyball, soccer, all those sports” (FG2)*, to appreciate the interactions they saw and to feel connected to others.


*“[The park is] just lovely, there’s always activity going on, there’s always somebody walking round the track round the field, or there’s somebody practising cricket, or there’s a little café there now, we’ve been there a couple of times… So I sit and watch all that interaction.” (FG3).*


#### 3.1.2. Holistic Health

Urban parks were a destination that had a positive effect on participants’ overall health. Participants discussed how visiting parks regularly and spending time in them was uplifting and influenced more than one aspect of their being; that is, physical, mental and emotional, and spiritual wellbeing. Participants readily spoke about the physical benefits of being physically active and that parks provided a place to be active, *“I go to a park for a bit of exercise” (FG1).* They noted that being active became more important the older one got, as one participant explained, *“If you don’t use it, you lose it” (P2).* While none of the participants in this study discussed the opportunity to participate in park-based group exercise, several described their experiences with park-based exercise equipment. The gym equipment was exciting and fun to use as well as good for physical activity.


*“There was four of us, and we all stopped [at these exercise machines in the park]. We were like three-year old’s, all having a go on these exercise machines. I found that quite interesting to find them.” (FG3).*


Walking in the park was perceived to improve mental and emotional wellbeing, *“I like to go for that walk, it’s good for your mental health” (FG4).* Parks provided a place to escape from stressors of daily life and enabled them to focus on different things, *“When I go out walking through a park, it stops me thinking about the things that I was thinking about at home. I think about completely different things” (FG3).* However, the environment of the park also allowed participants to just ‘be’, *“and you can still enjoy the sedentary part of being out in the park” (FG2).* Regardless of being active or just ‘being’ in the park, this environment helped participants overcoming feelings of stress, anxiety or depression. *“It’s very good for old people [to get out to the local park], otherwise we stay at home a lot, and you can get very lonely and depressed” (P4*).

Finally, participants used words such as peaceful, therapeutic, and tranquil to describe parks and noted that they provided a quiet place just to reflect. These attributes were good for the soul and for restoring spiritual wellbeing, *“I find that walking in the bush in fact, and walking through the parks and that sort of thing is—I think it was Psalm 23, ‘it restoreth my soul’” (FG2).*

### 3.2. Park Use Considerations

The practical and logistical factors participants considered when determining whether to use a specific park or to go to a park at all were evident. Furthermore, participants did not necessarily feel ‘safe’ or ‘invited’ at their local park. Sub-themes of ‘Travel logistics’, ‘Park amenities’ and ‘Not a space designed for me’ captured their experiences.

#### 3.2.1. Travel Logistics

Two aspects of the ‘Travel logistics’ were evident in participant discussion; both aspects of the Accessible Journey concept. One captured the favorable logistics that enhanced the decision to go to the park. Participants discussed how they could walk, *“I’m twenty minutes one way from it [the local park], so it’s a perfect walking time for me. That’s a forty minute walk each morning” (FG3).* Participants were grateful that there were still parks nearby that they could easily get to, *“Thank goodness, there are still plenty of places where you don’t have to walk very far [to get to a park]” (FG2).* For some, the walk to the park was part of the experience of visiting the park because the journey was an opportunity to connect with their neighborhood, *“The journey on the way is part of it as well. You see what’s going on … who’s painted their house and who hasn’t” (FG3).* Others however, made decisions based on drivability, *“We love parks that we can drive to” (FG2).* Participants also talked about the proximity of the park to other community amenities, which meant more activities were achieved within a day without the need for additional travel. This made visiting a park more logistically appealing, *“[The park is] within walking distance...The library [is also close] and also [we] go down to the playground at the beach” (FG2).*

Conversely, participants also discussed a number of issues related to the inaccessibility of parks due to distance and the terrain of the local area. One participant shared a story of wishing to take her mother to the park as this was an environment they both shared positive memories of, *“I had to actually get a taxi” (FG3).* When parks were difficult to get to participants reported feeling dependent on others for transport and for safety (due to the increased distance from home). As this was not an emotion participants enjoyed experiencing, decisions to forgo a park visit were sometimes made.

#### 3.2.2. Park Amenities

Amenities within the park made the option of visiting a park more viable. While participants did discuss the availability of seated refreshment areas (such as cafés) as important, the quantity and location of bathrooms, seating/park benches, and shelter within park environments were predominantly discussed. Bathrooms/toilets were felt to be very important; they needed to be clean and close by other main attractions, *“Elderly people tend to go to the toilet quite often, so you need a toilet close by. And clean toilets” (P2).* Yet bathrooms were often few and far between. Although said slightly flippantly, one participant suggested that a taxi was almost necessary to get from one side of the park to the bathrooms, *“That’s another problem, it’s too far—if you’re on the other side of the park, you need to ring a taxi. To come all the way over where the toilet is” (FG1).* The point being that the location and frequency of bathrooms was often insufficient for the participants’ needs. Interestingly, none of the participants mentioned anything specific about the quality of accessible bathrooms. However, finding the accessible bathroom padlocked closed was immensely frustrating.

Participants viewed seating as a *“need”.* Seating provided vital rest stops, places to gather thoughts, and vantage points to oversee grandchildren playing, *“I go to the playground with my grandchildren. There’s a seat there [that I can use]” (FG2*). Despite the need for seating, there were reports of *“nowhere to sit”.* Limited seating opportunities curtailed exploration involving greater distances from the point of park entrance as there was nowhere to recover.

The importance of shelter in providing *“a space out of the sun”* and *“cover”* was also identified. This was particularly pertinent around areas of large open expanse, such as playgrounds. Some participants mentioned that good quality shelter was non-existent at the park they frequented. *“[The park] could do with a little bit more shelter, particularly on the hot summer days” (FG3).*

For all participants, information about a park’s amenities, parking, bathrooms and consideration of accessibility design helped make park use decisions easier. However, participants noted that information could be hard to find, not helpful once at the park and not always available in accessible formats. For some participants, the energy required to visit a park, which then turned out not to be suitable, was a risk not worth taking, *“I like to find out if there’s a toilet first before I go” (FG2).*

#### 3.2.3. Not a Space Designed for Me

This sub-theme reflected the participants’ feelings of being excluded from parks. Reasons included the lack of accessible amenities, fear for physical safety, the equipment only being for young people and the park not being culturally accessible or appealing. Perceptions that parks had not been consciously designed to include older adults with disability were consistent throughout this sub-theme, *“It doesn’t cater for anybody with disabilities” (P2).*

Participants felt a lack of appropriate and accessible amenities and inaccessible environments in parks limited the choice of which parks they could go to. This included factors such as curb heights; path width, gradient and quality; lighting; and inclusive picnic tables. Participants suggested that there was a lack of *“consideration of design that included people with disability” (FG2).*

Several of these factors were identified as key safety concerns, with feeling unsafe linked particularly with physical safety concerns. The presence of physical hazards within the park environment made participants perceive that parks were less suitable for them. Path quality and gradient was a strong deterrent to park use, with issues of *“uneven paths and obstacles” (FG1)* and *“too many curbs you’ve got to step over, because it’s so easy to fall over now” (FG2)* reported. Some parks were even identified as *“too dangerous”* to access, *“That’s put us off some of the walks, you know, the steps- the board in the front and the thing in the back, it’s dangerous” (FG3).* Insufficient lighting in parks was also regarded as a physical safety issue. While lighting provided *“A bit of comfort” (FG3)* it was viewed as *“Important in any park, especially at night” (FG3)* as it improved visibility of upcoming hazards.

Participants also expressed the desire to use park ‘play’ equipment, including swings, slides or exercise machines because *“You don’t want to lose your youth” (FG1).* Play equipment was identified as an opportunity to step out of their comfort zone and challenge themselves physically (for example, with balance) and psychologically. Playing was perceived to help the participants stay in touch with their younger self, *“Just something about being a child again, I guess. I always go and sit on a swing” (FG3).* However, participants were nervous about using the play equipment; they intimated that play equipment was not for older adults let alone older adults with a disability, *“I’d say it is a young people’s park. Yeah, you don’t see many elderly there” (P1).* This made them self-conscious and consequently some participants even reported that they completely avoided playgrounds and only used them *“if there’s not too many other people [around]” (FG1).* A few participants noted that some parks visibly indicated with signage who was ‘allowed’ to play on the equipment, but this communication was not accessible in other formats. If signage existed, rarely did it provide older adults with disability with explicit permission to play.

Finally, the cultural relevance of parks was also discussed. Participants of Māori and Pacific descent suggested that the parks were not designed for people like them. They suggested that their cultures used parks differently to the dominant culture (NZ European); therefore, the current design made the spaces feel like they were unfriendly and exclusive. They acknowledged that traditional English designed garden beds were beautiful but wished to see some of the beautiful green space elements from their own culture also included. They suggested including Māori and Pacific art works, naming areas in their language, and including natural elements that were identifiably from the Pacific such as shells, and specific color combinations.


*“I want something [about the park that is culturally inviting]. Where we [can] go for some exercise. … We got different interests … you hardly see any [Pacific Island] people in the park. It must be the way they design it, you know, we’re not really interested in the way they [have] design [ed some parts of] the park.” (FG1).*


Participants strongly suggested that to make parks more inviting for people from different cultures, then input into the design of parks would be needed from people with disability of different cultures living in the local area.


*“If you really want to make [parks] more user friendly in this area, you’d want input about the design of the place from people [knowledgeable about their] culture… If you’re going to put Pacific Island culture into [park design] it would make such a difference, cause people would feel they have input in and have put their hand on things. It’s all about that with culture. They like to have something [that] respects and represents their culture, and have cultural events and items that recognise that [in the park]. It’s really missing in that [park].” (FG1).*


In summary, if the park environment was inaccessible and unsafe, participants found it necessary to seek additional support thus increasing dependence. Increasing dependence on others was not perceived to be a good health outcome from park use, *“I don’t want to be a burden to the family, especially my wife. Because that means it’s her that will push me around most of the time” (FG2).* Furthermore, if the environment and amenities in a park were inaccessible, then the park was perceived to not originally meant to have included them. The park became uninviting and exclusive, *“There’s just very little to entice elderly people to try” (FG3).* This resulted in some of the older adults with disability suggesting the parks that were in their area were not for them.


*“Well I walk most mornings, down to our local park. I don’t stop in the park cause there’s nothing for me to stop for.” (FG3).*


## 4. Discussion

The aim of this qualitative study was to explore the experience of park use and accessibility among older adults with disability. Two major themes were identified. These were “Enticing” and “Park use considerations”. The theme of “Enticing” captured how park use facilitated connection with the ‘self’ and with others, and contributed to overall wellbeing of older adults with disability. “Park use considerations” presented perceptions of accessibility and inclusion both on the journey to the park and within the park environment.

Participants were enticed to use parks because of the opportunity to connect and to improve their health. Being connected with others is an important aspect of human health and wellbeing [45,46] that is, the two themes presented within “Enticing” are interlinked concepts. Previous studies have similarly shown that urban parks provide opportunities for people to connect with others in the natural environment [47], and people who spend time in a natural environment are more likely to report good health and wellbeing [48]. White et al. [48] self-report survey study with 19,806 people reported those who were outside in nature for between 120–180 min in the previous week were more likely to report good health (OR = 1.59 [95% CI: 1.31–1.92]) and high wellbeing (OR= 1.23 [95% CI: 1.08–1.40]) than those who had no contact with the natural environment. This link between time in nature and good health and high wellbeing was similar across groups including older adults and those with long-term conditions, after controlling for factors of amount of residential greenspace, neighborhood and individual factors. While the review by Saitta, Devan, Boland and Perry [28] found two previous studies discussing positive social connectivity outcomes from urban park use, specifically in people with disability, our research is the first to explicitly present the interlinked importance of urban parks for connection and wellbeing in older adults with disability. Furthermore, in our study connection referred not only to others but also to ‘self’ and ‘self as a part of nature’.

Results from our study demonstrated perceived physical, psychological and social benefits from park use by older adults with disability. Physical activity is recognized as one means for maintaining health and this is particularly important for older adults and people with disability, as less than 50% of these populations meet daily physical activity requirements [4,49,50]. Participants in our study predominantly described walking by themselves or with others to improve their physical health, thus the quality of the paths was instrumental. However, a few also discussed the value of park-based exercise equipment. These findings agree with Veitch et al. [26] who found that three core elements (paths, park-based group exercise and exercise equipment) facilitated physical activity. While New Zealand largely has a temperate climate, it can experience −10 degrees Celsius in winter and sub-tropical temperatures in Summer. These variations in temperatures may be one reason why none of our participants described participating in organized park-based group exercise opportunities. Research does suggest that older adults involved in organized exercise programs are more likely to reach the recommended levels of physical activity and report higher self-efficacy [51]. Therefore, the development and promotion of age and ability considered organized park-based exercise opportunities might be a viable option in the warmer and sunnier months.

The attraction of a natural environment featured as an incentive for park use amongst the participants, relieving psychological distress and escaping from surrounding urbanization. Previous research has likewise found in a range of populations, that green space/nature promotes feelings of peacefulness and serenity [52,53,54,55]. Furthermore, visually pleasing aspects such as flowers, gardens and birds are known to play a part in reducing stress and facilitating relaxation [56,57,58]. For our participants the presence of these features, the excitation of all the senses and having a place to ‘be’ was a major factor in the choice of park to visit.

Our participants were interested in engaging in park play, particularly playing in the playground. Yet, they felt inhibited by a perceived lack of suitable equipment and the perception that equipment was not for their use. When a desire to play in the park and or playground was indulged, a feeling of ‘youthfulness’ was identified. They felt nostalgic as they remembered past experiences but also joyful as they felt young again. Therefore, minimizing the perceived need for “permission to play” may be achieved by signposting ‘Leisure and Intergenerational Seniors’ playgrounds [59], with playground designs targeting older adults’ coordination, motor skills, and memory functions. The design of these types of play areas also facilitates social interaction between older adults and younger age groups, increasing the number of age-appropriate spaces [59]. Just as participation in park related sport and leisure activities in older adults generates a sense of empowerment and identity, with improved life satisfaction and self-esteem [60], playground participation could foster risk-taking and self-confidence.

In our study, numerous factors limited the number of parks older adults with disability could access. These factors included a non-accessible journey, an environment which promoted dependence on others or posed a perceived risk, and a perception of park age, disability and cultural exclusion. In contrast to Costigan et al. [61] study which ranked, in people over the age of 60, the natural appeal and availability of seating as more important than the ease of getting to the park, our study found that difficulty getting to the park was a significant limiting factor for park use in older adults with disability. With respect to an accessible park environment, the number and location of amenities; seating, inclusive picnic areas and accessible bathrooms; navigable paths; and accessible information on park amenities determined whether participants in our study would visit a park at all. A previous study including individuals with disability, found that safety, ease of accessibility and observable maintenance were considered important components for park facilities [62]. Therefore, while older adults may value specific features for physical activity, other features were more important for social connection [26]. The results from our study are more in line with Kaczynski et al. [63] who found that parks with more facilities/amenities are preferred by older adults, compared to parks with more appealing environments but fewer amenities. These discrepancies may be related to the (dis)ability of the different cohorts across these three studies, with some older adults with disability requiring more resources to ensure inclusion in the park environment.

Participants identified park features that created a falls risk. These included: curbs; poor lighting; neglected stairs with no element (nosing or paint) to identify the lip; and paths with uneven surfaces, steep gradients and no markings. Fear of falling leads to decreased participation in outdoor activities, and the purported health benefits of park use could be lost [64]. Experiencing a fall can be a traumatic experience for older adults, leading to potential distress, embarrassment, and injury [65]. Furthermore, the perception of the environment affects risk of falls, with previous research finding people who perceived an environment to be favorable were less likely to fall [66].

The participants’ concerns about park amenities and risk of falls suggest that further consideration regarding the frequency and location of wheelchair-accessible bathrooms across the park environment, number of disability car parking spaces, quality and maintenance of footpaths and curbs, and seating at various heights is required [67]. As older adults, never mind older adults with disability, are vulnerable to equity and environmental barriers to participation in outdoor activities [62,68], we recommend that further consideration of the balance of park amenities and quality of amenities to park footprint may be required. Physical and psychological impairments secondary to falling may be minimized through adequate maintenance and co-design of safe park areas.

Our results indicated that park amenities were critical for park choice and enjoyment for the participants in our study. When these needs were unmet, the emotional and physical cost of going to a park outweighs the benefits. Therefore, the availability of accessible information on a park’s amenities could determine park use. Information regarding the location and availability of amenities has been shown to be important to park-goers who are older, providing a way to successfully manage their needs during park visits [69]. Our results supported this, with participants noting that a lack of knowledge about the location of facilities within a park negatively affected their experience. However, a point of difference from the results of our study was the need for information in a variety of formats to ensure all older adults with disability could access this information. While information about amenities at destination type parks is often available online, information about amenities at other types of parks are not always available this way. Park planners and designers along with council would need to bear in mind that older adults may have access, skill and efficacy issues in obtaining information via the internet [70]. Information about amenities needs to also be available, for example, in printed format, including large font, at places where people go frequently such as local libraries, or supermarkets or as signage at the entrance to parks, to maximize use of parks by older adults.

We purposively sampled participants from diverse ethnic backgrounds in this study. Our results show that different cultural groups use and value parks for various reasons; group social events, individual enjoyment or a free area for activities. Studies on the difference in park usage by various cultures have noted similar results, with minority groups (Pacific Islanders and African-Americans) participating in larger gatherings; while Caucasians tended to use parks individually or in pairs [71,72,73]. Participants in our study stated that cultural sensitivity should be evident in the park design, to facilitate park use for indigenous and/or minority groups. These results are important for two reasons. Firstly, this finding highlights the importance of acknowledging ‘identity’ concepts. From the social disability model perspective, disability is a culture [74,75,76]. For many people with disability their uniqueness is a part of their identity, it is society which places limitations on participation [75]. However, Māori and Pacific people generally prefer to identify as Māori and Pacific first [77,78]. For example, in te ao Māori (the Māori world), disability is just one aspect amongst many of the collective, cyclical experiences of life [77]. Thus, even if a park was disability accessible, if it was not also culturally responsive, Māori and Pacific participants with disability still felt excluded. Intersectionality, is the second reason why our result of cultural sensitivity is important. As per many other countries, minority and indigenous groups in New Zealand experience health inequities [34] due to avoidable, unfair, and unjust structural inequalities in multiple domains or systems of the social determinants of health [79]. People with disability similarly experience health inequities in New Zealand [34]. Intersectionality refers to the multiplicative effect from discrimination arising from the presence of two or more identities which are marginalized. In other words, health outcomes for Māori and Pacific people with disability are worse than for Māori and Pacific people without disability [34]. Therefore, as parks are so important for maintaining and improving health [19,20,28], parks that are not inviting to Māori and Pacific people with disability, arguably perpetuate intersectional discrimination. Addressing cultural diversity in park design could maximize access for a wider demographic to gain benefit from visiting urban parks.

Park planners and developers need an increased awareness of what older adults with disability seek in parks, and why certain facilities are necessary or desirable. Community collaboration is beneficial, as it “offers a dynamic, process-based mechanism for resolving planning issues,” [80]. Results from our study indicated that participants would relish the opportunity to have input into park design. They had numerous suggestions, based on ‘lived experience’ of disability. However, such a process requires commitment from councils and planners to ensure meaningful collaboration via on-going co-design to ensure accessible urban parks. Indeed, in New Zealand, consultation *only* at the end of the design process would renege on several Te Tiriti o Waitangi principles: Tino rangatiratanga, which guarantees self-determination in the design, and delivery of services; Active protection, which requires the Crown to act, to the fullest extent practicable, to achieve equitable health outcomes for Māori; and Partnership, which requires the Crown and Māori to work in partnership in the governance, design, delivery, and monitoring of services [81]. Furthermore, New Zealand, as per many other countries, is a signatory on the United Nations Convention on the Rights of Persons with Disabilities [39]. A process of co-design, specifically including local people with disability, as opposed to ad-hoc advocacy and consultancy from advisory groups and non-government organizations, would ensure that many Articles of the Convention are upheld.

### 4.1. Implications

The results from this study have implications for health professionals, policy makers and city councils. For health professionals working with older adults with disability, it is essential to be aware of potential safety and access-related issues that might inhibit positive health outcomes from park use. However, when appropriate, health professionals might also more readily consider ‘prescribing’ urban park use to maintain and improve biopsychosocial health outcomes in older adults with disability. This is particularly pertinent as parks are typically low cost or free and are therefore a viable alternative to group based or individual exercise programs held indoors or at a gym. For park planners, policy makers and city councils Table 4 provides a synthesis of participant concerns and our recommendations based on the study results.

However, these recommendations are only relevant to the parks the participants in this study have experienced. Therefore, the key implication is the need for improved collaboration and engagement with people of all ages, abilities and cultures to maximize park use and ensure park derived health benefits for all. To remove tokenistic end-stage consultation, a participatory park co-design process with people with disability (who have diverse cultural backgrounds), akin to that recommended for the design of health service delivery, should be considered. Co-design ensures meaningful involvement of the end-user at the beginning, and throughout the design process, to ensure the product or health service (park) will be usable [82]. We suggest that co-design is appropriate as parks are related to physical activity, an important public health consideration [20]. Advocacy for park accessibility will be further improved when the number of park planners and designers with disability is representative of disability in the general population.

### 4.2. Strengths and Limitations

While many of our findings are consistent with previous research, in older adults, this study provides a unique insight into park use experiences from the perspective of older adults with disability. To date, limited literature has explored this topic. We present novel findings related to the enticing elements of urban parks but also the perception of exclusion as participants considered parks, which were not designed with them in mind. This affect was multiplicative for older adults with disability from an indigenous or minority culture. The results suggest a need for more diverse collaborative park co-design to help ensure equitable health outcomes and to reduce discrimination.

The General Inductive Approach for data analysis provided a strong framework to understand participants’ experiences and allowed for an initially inductive and secondary more latent (deductive) exploration of the topic. We used procedures to ensure trustworthiness of our data analysis, including purposive sampling for data collection, focus groups and individual interviews and independent parallel coding with an audit trail.

The results from this study are valid for the participants recruited. Accordingly, these results are not necessarily generalizable to other areas of New Zealand or the world, or to younger people with disability. However, our broad inclusion criteria allowed participants of different ethnicities (i.e., Māori and Pasifika), people living in areas of high deprivation across Greater Wellington region to contribute, thereby maximizing the diversity of participant views [83].

## 5. Conclusions

This study explored experiences of parks, including accessibility, among older adults with disability. Older adults with disability perceived urban parks to be enticing as they enabled connection to ‘self’ and others, and health benefits. Park use was determined by considerations, such as the ability to independently get to a park, the accessibility of the park environment including the location and number of amenities, park information available in accessible formats, and perceived inclusiveness of the park environment. Whilst enticing, parks were not perceived to be designed for older adults with disability and were therefore exclusive. Park co-design between park planners and people with disability may provide one means of improving accessibility and park usability and thus park participation by older adults with disability.

## Figures and Tables

**Table 1 ijerph-18-00552-t001:** Semi-structured focus group and interview guide: Key areas for exploring.

**Section One:**
Introductions and greeting. How has your day been?
**Section Two:**
Tell us about the parks and playgrounds you visit How/Why do you go to the park? Who do you go to the park with? Why do you come to this park? (Clarify use of amenities) How has your use of parks and playgrounds changed over time? Has your impairment changed how and why you use a park? Questions around barriers and facilitators of using parks? What are your favorite things to do in this park? Why are these favorite things? Tell me about the things that are not your favorite things to do? How would you design a park for you?

For all questions, participants were offered the opportunity to provide an example, describing the situation in more detail.

**Table 2 ijerph-18-00552-t002:** Summary of participant demographic information.

Demographic Descriptions	*n* (%)
**Age Groups**	
Aged 55 to 64	2 (11.8%)
Aged 65 to 79	8 (47.0%)
Aged 80 and older	5 (29.4%)
Undisclosed	2 (11.8%)
**Gender**	
Male	9 (52.9%)
Female	5 (29.4%)
Undisclosed	3 (17.7%)
**Ethnicity**	
NZ European	13 (76.4%)
Māori	2 (11.8%)
Pacific	2 (11.8%)
**Education**	
High School (Secondary)	7 (41.2%)
Polytechnic or College	4 (23.5%)
University	6 (35.3%)
**Impairment that limits daily activities ***	
Hearing	1 (5.0%)
Neurological	1 (5.0%)
Vision	2 (10.0%)
Physical/mobility	13 (65.0%)
Psychiatric/physiological	2 (10.0%)
Other	1 (5.0%)
**Geographic area of the region participants reside**	
South	6
North East	5
North West	6

* Note some participants reported multiple impairments, therefore have been counted more than once.

**Table 3 ijerph-18-00552-t003:** Themes constructed from interviews with people living with disability about urban park use and accessibility.

Themes	Sub-Themes	Descriptions
Enticing	Connecting of ‘self’	Urban parks connect people to ones ‘self’, to nature and to other people (physically and vicariously).
	Holistic health	Parks are uplifting and positively affect more than one aspect of biopsychosocial and spiritual health and wellbeing.
Park use considerations	Travel logistics	Both favorable and unfavorable aspects affect the decision to go to the park.
	Park amenities	Amenities and safety within the park, and accessible information about park amenities made parks more logistically viable to visit.
	Not a space designed for me	Designers need to collaboratively co-design urban parks with diverse groups of people in the community to ensure the space is flexible and inclusive and meets the needs of older adults with disability.

**Table 4 ijerph-18-00552-t004:** Synthesis of accessibility considerations by older adults with disability.

Consideration	Recommendations for Improving Park Accessibility for Older Adults with Disability
Toilets	Has accessible toilets (i.e., more than one).
	Located close to main attraction(s).
	Placed at multiple locations on main arterial routes.
Seating and picnic areas.	Has seating at various heights with and without handrails.
	Has seating frequently along main arterial routes.
	Has seating which enables a wheelchair user to feel included
	Has picnic tables which enable a wheelchair user to be included.
	Has picnic tables over solid surface to support wheelchair at the table.
Paths and stairs	Even surfaces with clear markings.
	Meets minimum standard * curb height.
	Meets minimum standard * path gradient.
	Meets minimum standard * ramp gradient.
	Use of nosing or paint identify to lips and edges.
Lighting and shade	Good lighting for main routes for park use at night
	Shade protection for main attraction areas.
Equipment (play and exercise)	Clearly signposts equipment for older adults in accessible formats.
	Provide permission to play in the playground in accessible formats.
	Provide playgrounds and equipment that are accessible for a variety of abilities.
Park information	Provide information on park amenities and their locations, and park activities in a variety of accessible formats and mediums.
	Ensure park information is available not only at the park.
Culture	Provide culturally relevant art features and sculptures for the neighborhood population.
	Landscape and design elements of the park sensitive to the neighborhood population.

* Minimum acceptable standards for accessibility can differ by country.

## Data Availability

The data presented in this study are available on request from the corresponding author. The data are not publicly available due to ethical restrictions.

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
