# Peer review of "“Enticing” but Not Necessarily a “Space Designed for Me”: Experiences of Urban Park Use by Older Adults with Disability"

_ijerph, 2021, doi:10.3390/ijerph18020552_

Round 1

Reviewer 1 Report

This article presents experiences of urban park use by older adults. This is a very interesting topic considering the target audience and the analysis tools used. Older adults are often a traditionally forgotten vulnerable group.

I congratulate the authors because I believe that this proposal is quite interesting and useful for older people, especially with disabilities. The following comments are intended to help improve the quality of this work.

- To have a better understanding of the article, it is recommended to include a paragraph at the end of the introduction section describing the structure of the article.

- It is suggested that the Consolidated Criteria for Reporting Qualitative Research (COREQ) guide be described in greater detail to understand its application.

- In line 143 the text is not understood: … transcribe 3rd transcribed…

- The conclusions of the study are not clear and it is recommended to expand them.

- Check all the website references since in several of them there are no URLs.

- Finally, it is recommended to review English and grammar throughout the document.

Reviewer 2 Report

The article is well structured and well-written, and it has some new information. At least for me, it was an eye-opener that playgrounds could have multiple functions by being attractive also for elderly people. The introduction is focused and the method is relevant and accurately described.

My only concern is the result chapter, which I don't find logically structured, since the two headlines "enticing" and "practicalities" are not complementary. I therefore suggest either use "repelling" or something synonymous together with "enticing" or divide "practicalities" into two section; one on accessibility and one on amenities. Furthermore, I miss the safety issues in the result chapter and suggest moving the paragraph on fall risks (from row 469) from discussion to results and supplemented with some quotations.  

Not understood: row 425-428. Please explain what does the temperate climate has to do with participating in exercise groups?

Reviewer 3 Report

The paper deals with the theme of health benefits (physical, social and psychological wellbeing) related to the experience of urban parks use by older adults with disability. It is mostly based on a survey of interviews to older adults randomly selected, who were asked about their experience of using an urban park.

I find the results of this questionnaire to be quite obvious and the scientific goal of this research is not clear. It is amply proven by the scientific literature (and also by people's common understanding) that urban parks play an important role in the well-being of cities and citizens.

The question of the accessibility of urban parks, understood as inclusive places and designed according to the principle of "design for all", is also widely known. The answers reported by the questionnaire on personal experiences of use of a park in New Zealand only verify things that are obvious and the reprocessing of these data does not lead to any novel findings.

I think that this research lacks a reworking of the data useful to understand what are the performance requirements that a park should have to promote the wellbeing of adults with disabilities.

How should the public space of urban parks be shaped to promote socialization? What types of activities make a park attractive? What are the exclusionary elements that are perceived by park users? Etc. If no usable data in terms of performance are obtained from the interviews, the recommendation given to urban designers to pay attention to disadvantaged users when designing an urban park, remain generic.

I think that discussion and conclusions need to be improved. The title of the paper should also clarify the precise focus of the research.

Reviewer 4 Report

The current paper aimed to understand the experience of urban park use and accessibility by older adults living with disabilities for a better interpretation of facilitators and barriers in a region of New Zealand. The article is a qualitative study using the General Inductive Approach for the analysis of the interview undertaken with 17 older adults with self-reported disabilities. The paper is traceable and well organized. A good level of English was used in the paper, nevertheless, I suggest reading the article a few times more to control minor errors. The paper can be published after some minor touches are conducted. 

The Introduction section starts with the benefits of the physical activities, thereafter the section continues with the engagement of the physical activities by the older adults and the older adults with disabilities, the role of urban parks and green public spaces, the requirements of the physical activities by the older adults and the aim of the study. The introduction followed a good order of sub-topics, but I suggest adding the presentation of a paper structure at the end of the introduction. Also, accessibility is mentioned in many places in the article, therefore the definition of this term is required to add. Briefly, the definition could be ‘’ Accessibility of urban environments is an integrated measure of spatiotemporal availability, transport infrastructure, and land-use factors.’’ as studied in https://doi.org/10.1186/s12544-018-0334-4

The article followed with the materials and methods section with four subsections. The authors explained in order of the qualitative component of a sequential mixed-methods, the details of the interview & the participants, the interview questions & data collection, the General Inductive Approach as the method of the data analysis, the reason for the selected interview approach, and the reliability of the analysis. The section was sufficient and enough informative. Thereafter, the authors explained the results under the themes constructed from interviews with old people living with disabilities about urban park use and accessibility. The section was traceable and clear. Also, the results were discussed with the presentation of the limitations. Lastly, I suggest strengthening the conclusions section and the implications mentioned in the discussions can be integrated into the conclusions section with a deeper interpretation.

Round 2

Reviewer 3 Report

The paper has been improved with a table including some recommendations about accessibility of urban parks to older adults with disability. Conclusions have also been improved.

I respectfully disagree with some statements of the authors, but, anyway, the paper could be published in the present form.